# The Conservation of Long Intergenic Non-Coding RNAs and Their Response to *Verticillium dahliae* Infection in Cotton

**DOI:** 10.3390/ijms23158594

**Published:** 2022-08-02

**Authors:** Li Chen, Enhui Shen, Yunlei Zhao, Hongmei Wang, Iain Wilson, Qian-Hao Zhu

**Affiliations:** 1School of Life Sciences, Westlake University, Hangzhou 310024, China; chenli@westlake.edu.cn; 2Institute of Crop Science and Institute of Bioinformatics, Zhejiang University, Hangzhou 310058, China; ttxsenhui@gmail.com; 3State Key Laboratory of Cotton Biology, Institute of Cotton Research, Chinese Academy of Agricultural Sciences, Anyang 455000, China; yunleizhao2002@126.com (Y.Z.); aywhm@163.com (H.W.); 4CSIRO Agriculture and Food, Canberra, ACT 2601, Australia; iain.wilson@csiro.au

**Keywords:** long intergenic non-coding RNA, conservation, evolution, *Verticillium dahliae*, cotton

## Abstract

Long intergenic non-coding RNAs (lincRNAs) have been demonstrated to be vital regulators of diverse biological processes in both animals and plants. While many lincRNAs have been identified in cotton, we still know little about the repositories and conservativeness of lincRNAs in different cotton species or about their role in responding to biotic stresses. Here, by using publicly available RNA-seq datasets from diverse sources, including experiments of *Verticillium dahliae* (Vd) infection, we identified 24,425 and 17,713 lincRNAs, respectively, in *Gossypium hirsutum* (Ghr) and *G. barbadense* (Gba), the two cultivated allotetraploid cotton species, and 6933 and 5911 lincRNAs, respectively, in *G. arboreum* (Gar) and *G. raimondii* (Gra), the two extant diploid progenitors of the allotetraploid cotton. While closely related subgenomes, such as Ghr_At and Gba_At, tend to have more conserved lincRNAs, most lincRNAs are species-specific. The majority of the synthetic and transcribed lincRNAs (78.2%) have a one-to-one orthologous relationship between different (sub)genomes, although a few of them (0.7%) are retained in all (sub)genomes of the four species. The Vd responsiveness of lincRNAs seems to be positively associated with their conservation level. The major functionalities of the Vd-responsive lincRNAs seem to be largely conserved amongst Gra, Ghr, and Gba. Many Vd-responsive Ghr-lincRNAs overlap with Vd-responsive QTL, and several lincRNAs were predicted to be endogenous target mimicries of miR482/2118, with a pair being highly conserved between Ghr and Gba. On top of the confirmation of the feature characteristics of the lincRNAs previously reported in cotton and other species, our study provided new insights into the conservativeness and divergence of lincRNAs during cotton evolution and into the relationship between the conservativeness and Vd responsiveness of lincRNAs. The study also identified candidate lincRNAs with a potential role in disease response for functional characterization.

## 1. Introduction

Long non-coding RNAs (lncRNAs) are RNA molecules with a length longer than 200 base pairs (bp) and without protein-coding potential. Based on their genomic locality, lncRNAs are classified as long intergenic ncRNAs (lincRNAs), long intronic ncRNAs, and long anti-sense ncRNAs [1,2]. LncRNAs constitute an important component of the pervasiveness of genomic transcription [3] and function in wide ranges of biological processes in plants, including development and abiotic and biotic stress responses [4,5]. They achieve their functionality by modulating the transcription and/or translation of target genes in cis or in trans through diverse molecular mechanisms, such as chromatin remodeling, chromosome looping, regulation of mRNA splicing, and miRNA sponges [2,6].

Compared to protein-coding genes (PCGs) or mRNAs, generally, lncRNA transcripts are shorter, have fewer exons, are lowly expressed, show a more tissue-specific expression pattern, and are poorly evolutionarily conserved [7,8]. Comparative analyses of evolutionary trajectories of lncRNAs in both plants and animals showed that lncRNA transcripts evolve faster than PCGs and that lncRNA loci have a high turnover rate, leading to rapid gain and loss of lncRNA loci and drastic change in both the sequence and exon–intron composition of lncRNA transcripts. Consequently, the majority of lncRNAs are species-specific [9,10,11,12]. Transposable elements (TEs) contribute significantly to the origination and the rapid evolution of lncRNAs through excision or insertion as a result of active transposition [13,14]. For instance, approximately 65% of lncRNAs contain sequences homologous to the TEs in maize [15].

Cotton is the dominant crop producing natural fiber for the textile industry. Two diploid species (*Gossypium herbaceum* (A_1_) and *G. arboreum* (Gar, A_2_)) and two allotetraploid species (*G. hirsutum* (Ghr, AD_1_) and *G. barbadense* (Gba, AD_2_)) have been independently domesticated to produce long and spinnable fiber. It has been well-established that the allotetraploid species (AD-genome) derive from a progenitor species formed by a polyploidization event that occurred ~1.5–2 million years ago in the New World after hybridization between a Gar-like A-genome species and a D-genome American species close to *G. raimondii* (Gra, D_5_) [16,17]. Polyploidization is a driven force of plant evolution, rewiring the expression and interaction of both protein-coding and non-coding genes. Cotton is, thus, an ideal and powerful model for uncovering the evolutionary trajectories of genetic features, including lncRNAs. To investigate the dynamics of lncRNAs in the process of cotton polyploidization, a study has compared lncRNAs from Gar and Gra with those from Ghr and an artificial allotetraploid generated by crossing Gar and Gra [14]. Between Gar and Ghr or between Gra and Ghr, only ~10% of lincRNAs are conserved in terms of genomic location and expression. Similarly, between the artificial interspecific F_1_ and its two parents, only ~30% of lncRNAs are overlapping, meaning that many parental lncRNAs are silenced, and many new lncRNAs emerged in the artificial interspecific F_1_ due to the hybridization event [14]. These results imply dramatic reprogramming of transcriptome during the polyploidization process. The majority of Ghr-specific lncRNAs and the new lncRNAs identified in the artificial interspecific F_1_ seem to be derived from demethylated TEs, suggesting, similar to the observations in other species [13,18,19], that TEs are the major source of lineage-specific lncRNAs in cotton [14].

Given the potential functional significance of lncRNAs in cotton biology, several studies have conducted lncRNA identification in cotton using RNA-seq data generated from different tissues or plants subjected to different stress conditions [20,21,22,23,24,25,26,27]. The first large-scale discovery of cotton lncRNAs identified tens of thousands of lncRNAs using publicly available RNA-seq data generated from different cotton species and strand-specific RNA-seq data generated from 0–20 days-post-anthesis (DPA) ovule or fiber of Gba [23]. Based on co-expression analysis, the study also found several lncRNAs with a potential function in cotton fiber initiation and elongation [23]. In another experiment carried out by the same group, segregating progeny with different lint percentages derived from a cross between the fiberless (both lint and fuzz) mutant *Xu142fl* and its wild-type (WT) were used to uncover lncRNAs differentially expressed (DE) between the mutant and its WT during fiber initiation (0 DPA) or the early elongation stage (5 DPA). Several DE lncRNAs were functionally characterized using virus-induced gene silencing (VIGS) [28]. Comparison of the expression level of lncRNAs in 0 DPA ovule and 8 DPA fiber between the *Ligon-lintless-1* (*Li1*) mutant and its WT identified hundreds of DE lncRNAs, and two significantly down-regulated lncRNAs were found to be potential targets of ghr-miR2950 [22].

Many cotton lncRNAs with a potential role in tolerance of abiotic stress have also been identified [20,29], with several having been functionally characterized [25,30,31,32]. Knocking down a salt-responsive lncRNA, *GhlncRNA973*, in cotton by VIGS decreases the tolerance of cotton seedlings to salt stress, while its ectopic overexpression in *Arabidopsis* enhances tolerance to salt stress. *GhlncRNA973* is a predicted target of ghr-miR399 that potentially regulates the expression of *GhPHO2*, a homolog of *AtPHO2* (PHOSPHATE 2) involved in phosphate homeostasis [31]. The molecular mechanism by which *GhlncRNA973* regulates salt stress in cotton remains elusive. *GhlncRNA354*, an endogenous target mimic of miR160b that targets auxin response factors, was found to regulate salt tolerance and root growth [32]. *GhDAN1*, a drought-responsive lincRNA expressed in Ghr but not in Gar, could be a negative regulator of drought stress as silencing *GhDAN1* improves tolerance to drought stress. *GhDAN1* might achieve its regulatory role by binding to the AAAG motifs of the genes of the auxin-response pathways [30]. A more recent study applied VIGS to 111 lncRNAs and phenotyped the treated plants under four stress conditions, including drought, salt, heat, and cold. Approximately half of the lncRNAs were found to affect plant height (20), drought response (34), heat response (1), or cold response (5) [25].

Verticillium wilt caused by the soil-borne fungus *Verticillium dahliae* (*Vd*) is a destructive cotton disease worldwide. To know the potential role of lncRNAs in response to *Vd* infection, a study compared the expression profiles of lncRNAs that are conserved or non-conserved between Gba (*Vd* resistant) and Ghr (*Vd* susceptible) [26]. It was found that the proportion of *Vd*-responsive lncRNAs is higher among the non-conserved ones than among the conserved ones, and the non-conserved lncRNAs tend to have a higher expression level than the conserved ones. Two long anti-sense ncRNAs, *GhlncNAT-ANX2* and *GhlncNAT-RLP7*, generated from the *ANX2* and *RLP7* locus, respectively, were found to be negative regulators of *Vd* response as down-regulation of either of the two lincRNAs by VIGS enhances resistance to *Vd*. Both lncNATs probably achieve their functionality via the jasmonic acid (JA) pathway [26]. A more recent study identified 4277 DE lncRNAs based on comparison of transcriptomic data from *Vd* infected and mock root samples of a *Vd*-resistant Ghr cultivar [24]. For the DE lncRNAs, co-expressed trans-PCGs outnumber co-expressed cis-PCGs, implying that those DE lncRNAs could have a broad function by regulating the expression of PCGs not physically linked. The study also found that lncRNAs could be heavily involved in *Vd* response by regulating the JA pathway and demonstrated that the expression level of both *GhlncLOX3* and its trans-target *GhLOX3* is positively correlated with the *Vd*-resistance level of Ghr cultivars and that down-regulating *GhlncLOX3* in resistant Ghr cultivar by VIGS compromises the *Vd* resistance of the Ghr cultivar [24]. In addition, *lncRNA2* and *lncRNA7* were found to be negative and positive regulator of *Vd* resistance, respectively, by modulating genes involved in cell-wall development [33].

Despite the studies on identification of lncRNAs in different cotton species, the evolutionary dynamics of cotton lncRNAs before and after the polyploidization event remains largely elusive, as little is known about the birth and death, conservation, and diversification of lncRNAs during the evolutionary history of the *Gossypium* lineage, oe whether the conservation of lncRNAs is related to their functional conservation. To address these questions, in this study, we identified long intergenic non-coding RNAs (lincRNAs) in the two cultivated allotetraploid cotton species, i.e., Ghr and Gba, and their two extant diploid progenitors, i.e., Gar and Gra, and systematically compared the conservation of the identified lincRNAs, with a focus on those responding to *Vd* infection. We showed that while most lincRNAs are species-specific, the closely related subgenomes of Ghr and Gba tend to retain a higher proportion of highly conserved lincRNAs, and the *Vd* responsiveness of lincRNAs is positively linked to their conservation. We also identified several lincRNAs with a potential function in response to *Vd* infection or in regulation of miRNAs involved in modulating disease-resistance genes.

## 2. Results

### 2.1. Identification of lincRNAs in Diploid and Allotetraploid Cotton Species

To explore the repertoire and conservation of lincRNAs in the two cultivated allotetraploid cotton species [*G. hirsutum* (Ghr, AD_1_) and *G. barbadense* (Gba, AD_2_)] and their diploid progenitors [*G. arboreum* (A_2_) and *G. raimondii* (Gra, D_5_)], we collected a total of 610 published transcriptomic datasets generated from a diverse of tissues of the four species, including 84 datasets from the *Vd*-infection experiments (Figure 1A, Appendix A), because one of the major aims of this investigation was to uncover lincRNAs responding to *Vd* infection, so to provide candidates for further functional characterization.

Using the criteria specified in Materials and Methods (Appendix A), 6936, 5911, 25,425, and 17,713 lincRNAs were identified in Gar, Gra, Ghr, and Gba, respectively (Figure 1B). Between the two diploid species, while more lincRNAs were found in the A_2_ genome than in the D_5_ genome, the number of lincRNAs per dataset (112) was identical, suggesting that a larger genome size (~1.7 Gb of A_2_ vs. ~0.75 Gb of D_5_) does not necessary translate into having more lincRNAs, although the number of lincRNAs identified could also be impacted by sequencing depth. Between the two allotetraploid species, more lincRNAs were uncovered in Ghr than in Gba, probably due to Ghr having more datasets from not only more diverse tissues but also samples subjected to various stresses (Appendix A). Between the two subgenomes, more lincRNAs were found in the A_t_ subgenome than in the D_t_ subgenome in both Ghr and Gba, although the number of lincRNAs identified seem to be only weakly related to the size of subgenome (Figure 1B).

Compared to protein-coding genes (PCGs), lincRNAs are shorter (Figure 1D), are less likely to be expressed (Figure 1C), tend to be lowly expressed (Figure 1E), and are more tissue-specific (Figure 1F). In each species, most lincRNAs are located at ~1200 bp or ~15,000 bp away from their nearest PCGs (Figure 1G). The ~1200 bp peak is the distal promoter region of PCGs. Given the median distance (6862 bp in Ghr) and mean distance (24,302 bp in Ghr) of two neighboring PCGs (Appendix A) [34], enrichment of lincRNAs ~15,000 bp away from their nearest PCGs suggest that the lincRNAs in that region might be important in gene regulation, potentially acting as enhancers of their neighboring PCGs. However, compared to the three cultivated cotton species, the wild cotton species Gra seems to have a significantly higher percentage of expressed lincRNAs and to have its lincRNAs found at the nearby regions of PCGs, suggesting that domestication and artificial selection pressure of breeding practices may have shaped the expression dynamics and landscape of lincRNAs in the cultivated cotton species.

The Ghr lincRNAs were blasted against the full-length cDNAs generated from the leaf and root of two Ghr accessions (MCU-5 and Siokra 1–4) by PacBio SMRT (to be published separately). Approximately 47.1% of them had a hit (E-value ≤ 10^−6^). A total of 3278 lincRNAs (~27.4%) of those with a hit has ≥90% sequence similarity over ≥90% of the lincRNA length, meaning at least a quarter of the identified Ghr lincRNAs are highly confident ones.

### 2.2. Conservation of lincRNAs in Diploid and Allotetraploid Cotton Species

Sequence conservation is associated with function conservation. We, thus, investigated sequence and genomic position conservation of the lincRNAs predicted in the four cotton species and grouped them into four types: syntenic and transcribed (ST), syntenic and allelically transcribed (SAT), positional conservation (PC), and genome-specific (GS; see Materials and Methods for the definition of each type) (Figure 2A,B; Appendix A). In all pairwise comparisons amongst the six (sub)genomes, generally the SAT and GS lincRNAs are the most abundant, suggesting frequent loss and birth of lincRNAs during the history of cotton evolution. In line with the genetic and evolutionary relationship amongst the cotton (sub)genomes, the (sub)genomes that are closely related have the highest proportion of ST lincRNAs and the least proportion of GS lincRNAs, such as the A_t_ or D_t_ subgenome of Ghr and Gba, while the (sub)genomes that are remotely related have the highest proportion of GS lincRNAs and the least proportion of ST lincRNAs, such as Gar and Gra (Figure 2C). However, to some extent, the proportion of ST and GS in the comparison between Ghr_A_t_ and Gar is lower and higher, respectively, than that in other comparisons of closely related (sub)genomes (Figure 2C), suggesting that, in terms of the lincRNAs in Gar and its two descendent A_t_ subgenomes (Ghr_A_t_ and Gba_A_t_), Ghr_A_t_ is more divergent than Gba_A_t_ when compared to their ancestral A genome donor Gar (A_2_).

The ST and PC lincRNAs were further classified into three families, one2one, one2many, and many2many, based on the number of homologs in each of their six (sub)genomes (Figure 2A). Most ST lincRNAs (78.2%) belong to the one2one family, while only less than 10% belong to the many2many family (Figure 2D), whereas for the PC lincRNAs, the number of lincRNAs in the three families is many2many (44.0%) > one2one (29.7%) > one2many (26.4%) (Figure 2E). In terms of the (sub)genome conservation of the ST lincRNAs, 62.3%, 24.0%, 7.3%, 3.9%, and 2.4% of the 8901 families (including 6962 one2one, 1202 one2many, and 737 many2many) are conserved in two, three, four, five, and six (sub)genomes, respectively (Figure 2D). Of the 6962 ono2one families, 48 (0.7%), 118 (1.7%), 332 (4.8%), 1527 (21.9%), and 4937 (70.9%) are conserved in six, five, four, three, and two (sub)genomes, respectively. However, the difference of the (sub)genome conservation of the PC lincRNAs (which belongs to 5780 families) seems not to be as significant as that of the ST lincRNAs (Figure 2E).

The ST lincRNAs of the 6962 one2one families are considered to originate from the common ancestor of Gar and Gra, as each of them has a pair of homologs in the A_2_ and D_5_ or in the A_t_ and D_t_ subgenomes. We classified them into four types based on their presence and absence in the two ancestral diploids (Gar and Gra), and whether the diploid homologous lincRNAs have been inherited to the corresponding (sub)genomes of the two allotetraploids (Ghr and Gba), to infer their evolution dynamics (Table 1). Type 1 (262 or 3.8%) contains the families with a homologous lincRNA identified in both Gar and Gra, and 18.3% of these lincRNAs are retained in both subgenomes of Ghr and Gba, representing the most conserved ones, whereas 81.7% of them lost in one or both subgenomes of Ghr and Gba after the divergence of Ghr and Gba. Type 2 (1349 or 19.4%) includes the families with the homologous lincRNA retained in Gar but lost in Gra after their divergence; 42.8% and 12.2% of these Gar lincRNAs are retained and lost in the A_t_ subgenome of Ghr and Gba, respectively; the remaining (45.1%) are retained in the A_t_ subgenome of Ghr or Gba. Type 3 (1397 or 20.1%) are the families with the homologous lincRNA retained only in Gra and lost in Gar after their divergence, with 36.1% and 16.7% of them being retained and lost in the D_t_ subgenome of Ghr and Gba, respectively; close to half of them (47.2%) being lost in the D_t_ subgenome of Ghr or Gba. Type 4 (3954 or 56.8%) includes the families with their both Gar and Gra homologous lincRNAs lost after the divergence of Ghr and Gba, as at least an A_t_ and a D_t_ homologous lincRNA were identified in Ghr and/or Gba (Table 1). These results indicate that only a tiny portion (0.7%) of the ST lincRNAs is very conserved and that the vast majority of the ST lincRNAs have lost their identity in at least one of the four species during the evolution trajectory of cotton.

Of the 48 most-conserved lincRNAs, 17 have hits matching PacBio full-length cDNAs from both the A_t_ and D_t_ subgenomes of Ghr (see an example in Appendix A), 9 and 3 have hits matching PacBio full-length cDNAs from the A_t_ and the D_t_ subgenome of Ghr, respectively, meaning that transcription of about half (47.9%, 46/96) of these conserved Ghr lincRNAs is supported by full-length cDNAs.

### 2.3. Relationship between Conservation of lincRNAs and Their Vd Responsiveness

We first identified lincRNAs responding to infection of *V. dahliae*, a fungal pathogen causing the Verticillium wilt disease in cotton, by comparing the changes of the expression level of all predicted lincRNAs in roots before and after *Vd* infection using transcriptomic data generated from *Vd*-inoculation experiments. Using the criteria presented in Materials and Methods, in Gar, between ~200 and ~350 lincRNAs were found to be differentially expressed (DE) in at least one of the three time points; overall, 96 were differentially expressed at both 24 and 48 hours post infection (hpi), and 32 lincRNAs were differentially expressed at all three time points (Figure 3A). In Gra, ~100 DE lincRNAs were identified at either 12 or 48 hpi, with 28 being common at both time points (Figure 3B). In Ghr, there are more DE lincRNAs at the early time points (6–12 hpi) than at the late time points (24–72 hpi) with 11 being common at all five time points (Figure 3C). More DE lincRNAs were identified in Gba than in the other three species, with the highest number observed at 24–48 hpi. Interestingly, about half (348) of the DE lincRNAs identified at 2 hpi were steadily differentially expressed at all other time points (Figure 3D). While most lincRNAs that were found to be commonly differentially expressed at multiple time points in one species have homologous lincRNAs in at least one of the other three species, the homologous lincRNAs were usually found to be not differentially expressed. That could be a result of functional divergence of the homologous lincRNAs or because of the use of different *Vd* isolates in different *Vd*-infection experiments. However, overall, the *Vd* responsiveness of lincRNAs is positively associated with their conservation, as the percentage of *Vd*-responsive lincRNAs is significantly higher in the ST lincRNAs than in the PC ones in all four comparisons (Figure 3E). This was further supported by the association between conservation of lincRNAs and their *Vd* responsiveness, i.e., the transcribed lincRNAs residing in the syntenic regions (ST and SAT) are more likely to be associated with *Vd* responsiveness than the GS lincRNAs (Figure 3F).

We further used the following criteria to stringently select a set of DE lincRNAs: (1) the expression level of the *Vd*-infected samples is all higher (up-regulated) or all lower (down-regulated) than that of the uninfected control; (2) for the upregulated candidates, the expression level of at least one *Vd*-infected sample is ≥5 TPM (transcripts per million sequenced reads); for the down-regulated candidates, the expression level of the uninfected control is ≥5 TPM; (3) the expression fold change caused by *Vd* infection is ≥2 in at least one *Vd*-infected sample. As a result, 8, 47, 81, and 380 DE lincRNAs were shortlisted in Gar, Gra, Ghr, and Gba, respectively. Approximately 50% of these DE lincRNAs are genome-specific ones, the remaining were found in at least two (sub)genomes and belong to one of the three families (Table 2). For the one2one family lincRNAs, as expected, more are conserved in two–three (sub)genomes than in four–six (sub)genomes. Despite one *Vd*-repressed Gra lincRNA and three *Vd*-induced Gba lincRNAs being conserved in all six (sub)genomes (Appendix A), their homologous lincRNAs in other (sub)genomes were not shortlisted by the stringent selection criteria, suggesting that the level of *Vd* responsiveness of the stringently selected lincRNAs tends to be species-dependent.

### 2.4. Subgenome Dominance of the Vd-Responsive lincRNAs

To know the effect of the polyploidization event on lincRNA characteristics, we compared the length, expression level, and specificity of lncRNAs from the two subgenomes in the two allotetraploid species, Ghr and Gba. The length difference of the A_t_ subgenome lincRNAs seems to be bigger than that of the D_t_ subgenome in both Ghr and Gba, while the median lincRNA length of the A_t_ subgenome is slightly longer than that of the D_t_ subgenome in Gba, but the median lincRNA length of the A_t_ and D_t_ subgenomes of Ghr seems to be similar (Figure 4A). A significant difference was observed for the maximum expression level of lincRNAs from the two subgenomes in both Ghr and Gba, and the lowest maximum expression level was observed for the Gar lincRNAs (Figure 4B). A significantly different tissue specificity was evident between the Ghr_A_t_ and Ghr_D_t_ subgenome lincRNAs but not between the Gba_A_t_ and Gba_D_t_ subgenome lincRNAs, despite lincRNAs of both Ghr and Gba seeming to have lost a certain level of tissue specificity compared to the lincRNAs in the two ancestral diploid cotton species, Gar and Gra (Figure 4C). These results indicate that certain features of the allotetraploid cotton lincRNAs have diverged from that of their ancestral diploid cotton, particularly from the ancestral A genome cotton, and that the difference between the two subgenomes in both Ghr and Gba is relatively small, comparing to the difference between diploid and allotetraploid cottons.

Regarding the number of *Vd*-responsive PCGs and lincRNAs, a similar number of up- or down-regulated PCGs were found in the two subgenomes of Ghr at each time point. For DE lincRNAs, while the number seems to be slightly higher in the A_t_ subgenome at 6 hpi, both the number and change trend at other time points seem to be similar between the two subgenomes (Figure 4D). Like Ghr, Gba has a very similar number of DE PCGs in the two subgenomes; however, more *Vd*-induced lincRNAs were found in the A_t_ subgenome at most time points; in contrast, in the D_t_ subgenome, the number of *Vd*-induced lincRNAs from 2–48 hpi are slightly less than that of the *Vd*-repressed lincRNAs (Figure 4E). These observations indicate that the two allotetraploid cottons may have different subgenome dominance regarding *Vd*-responsive lincRNAs.

### 2.5. Cis-Regulatory Role of the Vd-Responsive lincRNAs in Cotton

The functionality of lincRNAs is achieved by interacting with PCGs or other non-coding RNAs via diverse molecular mechanisms [4]. LincRNAs and their cis targets tend to be co- or reciprocally expressed, we, therefore, using the RNA-seq data from the *Vd*-infection experiments, analyzed the expression changes of the neighboring PCGs (one on each side) of the *Vd*-responsive lincRNAs found in the four cotton species, to identify those that were significantly induced or repressed in at least one time point following *Vd* infection. As a result, 208, 305, 228, and 903 such PCGs were found in Gar, Gra, Ghr, and Gba, respectively. These PCGs were subjected to GO-enrichment analysis. The PCGs from Gar were enriched with one biological process (BP) GO term (*p*-value < 0.001 applying to all) and one cellular component (CC) GO term. The Gra PCGs were enriched with one molecular-function (MF) GO term. The Ghr PCGs were enriched with two, one, and five biological process, cellular components, and molecular-function GO terms, respectively. The Gba PCGs were enriched with three and six biological process and molecular-function GO terms, respectively. Both GO terms enriched in Gar do not overlap with those of Gar, Ghr, and Gba; the MF term (iron ion binding) enriched in Gra is also enriched in Ghr and Gba; between Ghr and Gba, four terms overlap, including one BP and three MF terms (Figure 5; Appendix A). Based on these results, more biological functions or pathways seem to be regulated by lincRNAs in the allotetraploid cottons than in the diploid cottons under the conditions of *Vd* infection. While the cis targets of the *Vd*-responsive lincRNAs in Gra tend to be maintained in Ghr and Gba, those in Gar do not. Even between the two allotetraploid cottons, many *Vd*-responsive lincRNAs might regulate PCGs with different functionality, despite some of them that might regulate a set of PCGs with similar functionality.

GO-enrichment analysis was also done for the flanking PCGs of the ST and PC lincRNAs of the one2one family in the four species. Interestingly, no enriched GO was found for the flanking PCGs of both types of lincRNAs in Gar. For the flanking PCGs of the ST lincRNAs, in Gba, they are enriched with eight GO terms, and five of those are overlapping with those of the *Vd*-responsive lincRNAs; in Gra, the two enriched GO terms are different from that of *Vd*-responsive lincRNAs, and no enriched GO was found in Ghr. For the flanking PCGs of the PC lincRNAs, only a single enriched GO term was found in Ghr (Appendix A).

Together, the above results imply that, compared to the PCGs flanking overall lincRNAs, those flanking the *Vd*-responsive lincRNAs are quite unique, particularly in Gar, Gra, and Ghr, and that the regulatory functions of the *Vd*-responsive lincRNAs are largely conserved in cotton, particularly between Ghr and Gba, although some of their functions have diverged during the evolutionary history of cotton. 

### 2.6. Overlapping between Vd-Responsive lincRNAs and QTL

To further explore the potential function of the *Vd*-responsive lincRNAs identified in Ghr, we analyzed their overlapping with the reported quantitative trait loci (QTL) associated with *Vd* responsiveness. In total, 198 A_t_-subgenome and 269 D_t_-subgenome *Vd*-responsive lincRNAs were found to overlap with 55 A_t_-subgenome and 37 D_t_-subgenome QTL, respectively. Depending on the QTL size, each QTL contains 1 to 36 *Vd*-responsive lincRNAs. Of the 23 stringently selected *Vd*-responsive Ghr lincRNAs, 9 were found to overlap with eight QTL, with 1 in the A_t_ subgenome and 7 in the D_t_ subgenome (Appendix A). For the overlapping QTL regions with potential disease-response gene(s), their response to *Vd* infection might be contributed to by the potential disease response gene(s) or their interaction with lincRNAs. For instance, Ghrlnc.47594, a *Vd*-responsive lincRNA selected by the stringent criteria, is located at QTL-59, where Ghrlnc.47594 was found to be flanked by one disease-resistance gene and one leucine-rich repeat-containing gene. Nevertheless, the majority of the overlapping QTL do not contain a gene with a predicted and/or demonstrated function in disease response.

Of the 467 *Vd*-responsive Ghr lincRNAs overlapping with *Vd*-responsive QTL, 214 are Ghr-specific. Amongst these lincRNAs specific to Ghr, nine (Ghrlnc.3378, Ghrlnc.9353, Ghrlnc.9725, Ghrlnc.28968, Ghrlnc.29661, Ghrlnc.46363, Ghrlnc.63449, Ghrlnc.83232, and Ghrlnc.85022) are particularly of interest, as their expression levels were significantly changed (≥2 folds) upon *Vd* infection and highly expressed in *Vd*-infected samples (≥5 TPM, for the one that is up-regulated) or highly expressed in the uninfected control (≥5 TPM, for the eight that are down-regulated). These nine lincRNAs overlap with eight *Vd*-responsive QTL (Appendix A) and are good candidates for further investigation of their function in response to *Vd* infection. 

### 2.7. LincRNAs as Potential Target Mimicry of miR482/2118

LncRNAs that interact with miRNAs but cannot be cleaved by miRNAs are negative regulators of the miRNAs, termed as endogenous target mimicry (eTM) [35]. miR482 and miR2118 are negative post-transcriptional regulators of genes encoding nucleotide-binding leucine-rich repeat (NLR) proteins by targeting their conserved P-loop motif for transcript degradation or translational repression [36,37,38]. It is, thus, in our interest to know whether some of the cotton lincRNAs identified here are potential eTMs involved in regulation of miR482/2118-mediated modulation of NLRs and, consequently, disease-response outcomes. 

Using the methods presented in Materials and Methods, we predicted one, two, four, and eight lincRNAs to be potential eTMs of different isoforms of miR482 or miR2118 in Gar, Gra, Ghr, and Gba, respectively (Appendix A). Most lincRNAs can potentially interact with a single miR482 or miR2118 isoform, but a couple of them were found to be able to interact with two different miR482 isoforms, such as Ghrlnc.53204, which contains binding sites for both miR482a and miR482g. In one case (Ghrlnc.71386), the lincRNA was found to contain two binding sites of miR2118k. Importantly, we found a pair of homologous lincRNAs in Ghr (Ghrlnc.36832) and Gba (Gbalnc.31516) to be potential eTMs of miR2118e (Appendix A). Except for the intron found in Gbalnc.31516 but not in Ghrlnc.36832, these two lincRNAs are almost identical (Appendix A), implying their functional conservation. No matching sequence was found for these two lincRNAs in our PacBio full-length cDNA collections, likely due to the difference in the cotton accessions used and/or the low expression of the lincRNA, so it could not be detected by the sequencing depth used in generation of the full-length cDNAs. Nevertheless, three full-length cDNAs homologous to these two lincRNAs with 2–4 nucleotide polymorphisms at the miR2118e binding site were found (Appendix A), and mutation in these polymorphic site(s) could change them into potential eTMs of miR2118e.

## 3. Discussion

Eukaryote genomes are pervasively transcribed to generate many different types of non-coding RNAs, with lincRNAs being one of the major types. Studies in both plants and animals indicated that lincRNAs are usually lineage- or species-specific, and positional conservation is more common than sequence conservation, meaning that the lincRNAs that are broadly conserved in different species that share only short patch sequences [9,10,11,15]. For instance, while ~20% of rice lincRNAs have a detectable sequence similarity to the maize genomic sequences, only ~1% of them have a sequence similarity to the maize lincRNAs, in contrast, approximately a quarter of the rice and maize lincRNAs were found in the synteny blocks [10]. In a study comparing conservation of lncRNAs in three lineages, *Brassicaceae*, *Aethionemeae*, and *Cleomaceae*, it was found that, of the 6480 *Arabidopsis thaliana* lincRNAs [39], only 11 are conserved in *Aethionemeae* with 9 of them seeming to be transcribed, whereas 12 of the 39 lineage-specific lncRNAs are positionally conserved in at least one of the other lineages [40]. Even within the same *Brassicaceae* family, while only ~9% of *Brassica napus* lncRNAs showed sequence similarity with those from *A. thaliana*, ~44% of *B. napus* and *A. thaliana* lncRNAs are conserved by position [41]. Our results observed in the four closely related cotton species are consistent with these findings. In each pairwise (sub)genome comparison, despite the proportion of lincRNAs conserved by both sequence and position (those of ST) decreases with the increase in genetic distance of the (sub)genomes, a significant proportion of lincRNAs (those of SAT and PC) are conserved by position and not by sequence (Figure 2).

The ST lincRNAs are the most-conserved. Their retention in genetically distant (sub)genomes suggests that they might have undergone purification selection owing to their functional importance. The PC lincRNAs have diverged significantly in their sequences but retained their transcription. For these lincRNAs, their function, if any, might not be related to their sequences but to the transcription of the genomic locus containing the lincRNA. Although no such functionality has been reported for plant lincRNA, it has been reported in animals, such as *LncMyoD*, a lincRNA identified in mouse myoblast and regulating skeletal muscle differentiation, which showed no sequence conservation between mouse and human but was conserved by gene structure and function [42].

Position but not sequence conservation of lincRNAs indicates rapid turnover of lincRNA loci. Comparative studies of lncRNAs from 16 vertebrate species and the echinoid sea urchin found >70% of lncRNAs might have appeared in the past 50 million years, although no homologous lincRNAs are traceable in the conserved positions [9]. Similarly, in plants, despite 83–98% of *Citrus sinensis* lincRNAs have homologous sequences in eight closely related citrus genomes, only 16–29% of them were observed to be transcribed in the eight species [43]. The rapid turnover of lincRNAs is also evident in cotton, based on the presence and absence of the ST lincRNAs of the 6962 one2one family in the six (sub)genomes (Table 1). These lincRNA families are inferred to be originated before the divergence of the A and D genomes ~5 million years ago [17], rather than after their divergence, as at least a pair of A (A_2_ and A_t_) and D (D_5_ and D_t_) homologs were identified among the six (sub)genomes. Of these lincRNA families, only 0.7% have retained homologous lincRNAs in all six (sub)genomes, 19.4% and 20.1% lost the homolog in Gra (D_5_) and Gar (A_2_), respectively, and 56.8% lost the homologs in both Gar and Gra. While Gar is a domesticated and cultivated species and Gra is a wild species, the similar turnover rate observed in the two species suggests that domestication and artificial selection might have little impact on the evolution of these lincRNAs in the time period of ~5 million years. However, for both A_t_ (derived from A_2_) and D_t_ (derived from D_5_) homologs, their retention rate is ~2% higher in Ghr than in Gba (Table 1), implying a kind of species difference.

While it is still under debate whether the transcribed lincRNAs are functionally relevant, many lincRNAs have been demonstrated to be important regulators of diverse biological processes [44,45], including some preliminary results achieved in cotton [25,27,30,32,33]. Nevertheless, owing to the huge number of lincRNAs identified and their unique characteristics, such as low and tissue-specific expression, it is still a challenge to know which of them are functionally important, so they should be chosen for in-depth functional investigation to understand their underlying regulatory mechanism(s). Comparative genomics analysis of lincRNAs across related plant species, just like what we have done here, can provide practical clues for investigating function of lincRNAs, because like homologous PCGs, homologous lincRNAs are expected to have conserved function and, thus, are worthy of further study. 

One of the major goals of this study was to use comparative analysis to understand the relationship between conservation and the *Vd* responsiveness of cotton lincRNAs and to identify candidate lincRNAs for further functional characterization. We found that the lincRNAs conserved by sequence and/or position (ST, SAT, and PC) are more likely to be associated with responding to *Vd* infection than the genome-specific (GS) ones (Figure 3F). Compared to the PC lincRNAs, the ST lincRNAs have a much higher percentage of *Vd*-responsive ones (Figure 3E). The regulatory roles of the *Vd*-responsive lincRNAs in Gra, Ghr, and Gba seem to be largely conserved (Figure 5). In addition, a pair of lincRNAs highly conserved between Ghr and Gba were predicted to be potential eTMs of miR2118e, a negative regulator of several NLRs (Appendix A) [46]. These observations suggest that the function of cotton lincRNAs, if any, is likely to be related to their conservation level, although the level of response might be species-dependent.

A few studies have investigated *Vd*-responsive cotton lncRNAs [24,26,33]. One of those studies found more *Vd*-responsive lncRNAs in the D_t_ subgenome than in the A_t_ subgenome in both Ghr and Gba, and, consistently, slightly more *Vd*-responsive QTL were found in D_t_ than in A_t_ [26]. In contrast, we found more *Vd*-responsive lincRNAs in A_t_ than in D_t_ in both Ghr and Gba (Figure 4D,E). The discrepancy might be due to use of different cotton accessions and lncRNAs (all lncRNAs vs. lincRNAs) in the two studies. Despite the inconsistency, we also found more *Vd*-responsive QTL in D_t_ than in A_t_ for the stringently selected *Vd*-responsive lincRNAs. For the two lncRNAs (*lncRNA2* and *lncRNA7*) that have been demonstrated to be regulators of *Vd* resistance [33], although *lncRNA7* was not identified in this study, *lncRNA2* was identified, despite its expression change upon *Vd* infection being insignificant. These results suggest that the repertoire of *Vd*-responsive lncRNAs is not yet saturated and that the same lncRNA might respond differently to *Vd* infection due to different genetic background and/or different *Vd* isolates. More investigations involving diverse cotton accessions and pathogens (also different strains of the same pathogen) are, thus, required to have an in-depth understanding of the landscape of the cotton lncRNAs responding to disease infection and the function of lncRNAs in the interaction between cotton and pathogens. Ideally, such study could combine multiple strategies, such as strand-specific RNA-seq, SMRT full-length RNA-seq, and cap analysis of gene expression, in the integration of lncRNAs, so it can simultaneously investigate the alternative splicing, transcription start, termination site, expression level, and change of lncRNAs [47].

We used a comprehensive pipeline to identify and characterize lincRNAs from different cotton species and their conservation during cotton evolution, particularly the lincRNAs responding to infection of *Verticillium dahliae*. The pipeline is applicable to other plant species. While we have achieved what we aimed for, we also realize the limitation of the study, mainly the relatively small number of RNA-seq datasets from *Verticillium dahliae* inoculation experiments, which we hope can be overcome in the future when more such datasets are available.

## 4. Materials and Methods

### 4.1. Identification of lincRNAs in Diploid and Allotetraploid Cotton Species

In order to study evolutionary dynamics of cotton lncRNAs before and after the polyploidization event and their functional relevance to *Verticillium dahliae* (*Vd*) infection, we collected publicly available RNA-seq datasets generated from the two cultivated allotetraploid cotton species, *Gossypium hirsutum* (Ghr, AD_1_) and *G. barbadense* (Gba, AD_2_), and their extant diploid progenitors, *G. arboreum* (Gar, A_2_) and *G. raimondii* (Gra, D_5_), for lncRNA identification (Figure 1A; Appendix A). Given that most RNA-seq data are not strand-specific, we focused on lncRNAs in the intergenic regions, i.e., lincRNAs.

To identify lincRNAs, we firstly mapped RNA-seq reads from each species to its corresponding reference genome by HISTA2 [48] with the default parameters. The four cotton reference genomes, Ghr.TM-1.HAU_v1.1, Gba.AD2.HAU_v2_a1, Gar.CRI-updated_v1, and Gra.D5.JGI_v2_a2.1 [34,49,50], and their annotation files were downloaded from CottonGen (https://www.cottongen.org; accessed on 29 July 2022) [51]. The mapped reads of each sample/replicate were then assembled by the genome guided software StringTie (McKusick-Nathans Institute of Genetic Medicine, Baltimore, MD, USA) [52], and the assembled transcriptomes of the same cotton species were merged using the merge module in StringtTie (StringTie --merge). The assembled transcripts of each cotton species were filtered by length and protein coding potential using the following criteria to have the final set of lincRNAs. We removed transcripts with a length less than 200 bp and with the predicted shortest open reading frame (ORF) longer than 100 amino acids. We also used blastx to query the non-redundant protein sequences (NR) of NCBI (National Center for Biotechnology Information) and filtered out the transcripts with a hit using the cut-off threshold of E < 10^−10^. The remaining transcripts were further assessed by the coding potential test software CPC2 [53] and compared to the annotated protein-coding transcripts of the corresponding genome to remove those with a match. 

### 4.2. Identification of Homologous lincRNAs 

The lincRNAs from each of the two cultivated allotetraploid cotton, Ghr and Gba, were separated into two groups based on their subgenome (A_t_ and D_t_) origin, and homologous lincRNA analysis was done amongst the six (sub)genomes, i.e., Gar (A_2_), Gra (D_5_), Ghr_A_t_, Ghr_D_t_, Gba_A_t_, and Gba_D_t_.

We defined homologous lincRNAs based on pairwise comparison amongst the six (sub)genomes with three complementary approaches: sequence similarity by blastn, synteny and positional conservation based on flanking PCGs by MCScanX (https://github.com/wyp1125/MCScanx; accessed on 29 July 2022), and whole-genome alignment by Lastz (https://lastz.github.io/lastz/; accessed on 29 July 2022). Firstly, the repeat masked lincRNA sequences from each (sub)genome were reciprocally compared with each other by BLAST 2.4.0+ (-evalue 1 × 10^−5^ -num_threads 10 -max_target_seqs 1 -word_size 8 -strand plus -outfmt 6). LincRNA sequences from two (sub)genomes with an alignment E-value < 10^−5^ were considered to be the best hits and homologs [54]. Secondly, MCScanX [55] was used to identify syntenic lincRNAs in two (sub)genomes based on their flanking syntenic PCGs by pairwise comparison. We considered three PCGs at each side of a given lincRNA. A lincRNA that was found in two (sub)genomes, flanked by a minimum of one syntenic PCG on each side, and has a total of at least three syntenic PCGs, was defined as syntenic lincRNA [56]. Thirdly, lincRNAs of the query (sub)genome were lifted to the target one by UCSC LiftOver (https://genome.ucsc.edu/goldenPath/help/hgTracksHelp.html#Liftover; accessed on 29 July 2022) with the assistance of chain files, which were generated by whole-genome alignment using Lastz, were used to translate the syntenic regions from one (sub)genome to another, to find homologous lincRNA pairs between the two (sub)genomes. Lastly, for each comparison, the homologous lincRNAs identified by the three approaches were merged to have a final list of homologous lincRNAs. 

The homologous lincRNAs were assigned to four groups, syntenic and transcribed (ST), syntenic and allelically transcribed (SAT), positional conservation (PC), and genome-specific (GS), based on their conservation level. Syntenic and transcribed represents the situation where a pair of expressed homologous lincRNAs were identified in the syntenic position of two (sub)genomes. Syntenic and allelically transcribed represents the scenario where an expressed lincRNA was found in the syntenic region but only in one of the two (sub)genomes. Positional conservation means that a pair of expressed lincRNAs were identified in the syntenic region of two (sub)genomes, but their sequence homology has eroded to a point of insignificance. Genome-specific refers to those lincRNAs that were identified only in one (sub)genome and do not share position and sequence similarity with any lincRNA from another (sub)genome.

The ST and PC lincRNAs were further assigned into families. To that end, a lincRNA sequence similarity network was built to connect homologous lincRNAs from individual (sub)genomes. An unsupervised graph-cluster algorithm (MCL, https://micans.org/mcl/; accessed on 29 July 2022) was then used to identify lincRNA cluster within the constructed network with the parameter --abc -I 2.0. Each cluster of homologous lincRNAs was designated a lincRNA family that was then assigned to one of the three families: one-to-one (one2one), one-to-many (one2many), and many-to-many (many2many), based on the number of homologous lincRNA(s) in each of the individual (sub)genomes from which the lincRNA(s) were identified. If a lincRNA has only a single homologous copy in all (sub)genomes with the homologous lincRNA identified, the cluster contains these homologous lincRNAs was defined as a one2one family; if a lincRNA has a single copy in one (sub)genome and multiple homologs (≥2) in at least one of the other (sub)genomes, the cluster contains such homologous lincRNAs was defined as a one2many family; and if a lincRNA has multiple homologs (≥2) in all (sub)genomes from which the homologous lincRNAs were identified, the cluster containing the homologous lincRNAs was defined as a many2many family.

### 4.3. Quantification of lincRNA Expression and Identification of Vd-Responsive lincRNAs

The sequences of lincRNAs identified in each cotton species were merged with the annotated protein coding sequences of the same species to create a reference transcript dataset of the cotton species, which was indexed by Kallisto software (https://pachterlab.github.io/kallisto/about; accessed on 29 July 2022) with default parameters [57]. The expression level (TPM) of lincRNAs and PCGs in each sample was determined by Kallisto software based on the indexed transcripts. The average value of the replicated samples was used to represent the final expression level of lincRNAs and PCGs if replicates were available. For the RNA-seq datasets generated from *Vd*-infection experiments, the raw read count of lincRNAs and PCGs were used in identification of differentially expressed (DE) lincRNAs and PCGs (DEGs) by DESeq2 [58] with the criteria of q-value < 0.05 and |log2(FC)| ≥ 1. PCGs and lincRNAs with a TPM < 0.5 in at least one sample were eliminated in the DE analysis. The DE lincRNAs and DEGs identified were considered as *Vd*-responsive. Tissue specificity index (TSI) was used to quantify the expression specificity of lincRNAs and PCGs, and generated by using the methodology previously described [59]. The value of TSI is between 0 and 1, with 1 and 0 meaning tissue-specific and broadly expressed, respectively.

### 4.4. Analysis of Cis Targets of Vd-Responsive lincRNAs

LincRNAs regulate their target genes in cis and/or in trans. Here, our focus was on the potential cis targets of *Vd*-responsive lincRNAs. To identify such cis targets, the expression changes of the nearest left and right neighboring PCGs of *Vd*-responsive lincRNAs in response to *Vd* infection were calculated, and the PCGs co-differentially expressed with their neighboring *Vd*-responsive lincRNAs were considered as cis targets of the lincRNAs and were subjected to GO-enrichment analysis using GOseq [60]. GO analysis was also done for the flanking genes of the ST and PC lincRNAs of the one2one family.

### 4.5. Analysis of Association between Conservation of lincRNAs and Their Vd-Responsiveness

To investigate expressional and evolutionary dynamics of *Vd*-responsive lincRNAs, we first overlapped homologous lincRNAs, which were defined based on the pipeline described in Section 2.2 with differentially expressed lincRNAs in response to *Vd* infection, then used Fisher’s exact test (*p*-value) to judge whether homologous lincRNAs are overrepresented in *Vd*-responsive lincRNAs. Fisher’s exact test was also used to test the association between the *Vd*-responsive lincRNAs and their conservation, i.e., enrichment of the *Vd*-responsive lincRNAs in the ST, SAT, PC, and GS four types of lincRNAs.

### 4.6. Identification of Endogenous Target Mimicry of miR482/2118

We used in-house script, coded based on the criteria previously described [61], and psMimic [62] to identify endogenous target mimicry (eTM) of miR482 and miR2118 [46], the two miRNAs targeting genes encoding nucleotide-binding leucine-rich repeat receptors (NLRs).

## 5. Conclusions

By using RNA-seq datasets from diverse sources, including *Vd*-infection experiments, we identified a number of lincRNAs in the two allotetraploid cotton species (Ghr and Gba) and their ancestral diploids (Gar and Gra). Most lincRNAs are species-specific, despite many more conserved lincRNAs being found between the closely related subgenomes of Ghr and Gba than between the remotely related (sub)genomes. *Vd* responsiveness of lincRNAs is positively correlated with their conservation level, so many *Vd*-responsive Ghr-lincRNAs overlap with *Vd*-responsive QTL, and several lincRNAs were predicted to be eTMs of miR482/2118, including a pair highly conserved between Ghr and Gba. The results presented here verified the characteristics of plant lincRNAs as previously reported, expanded the repositories of cotton lincRNAs, shed new insights on the relationship between the conservation of lincRNAs and their *Vd*-responsiveness, and provided candidate lincRNAs for future functional investigation.

## Figures and Tables

**Figure 1 ijms-23-08594-f001:**
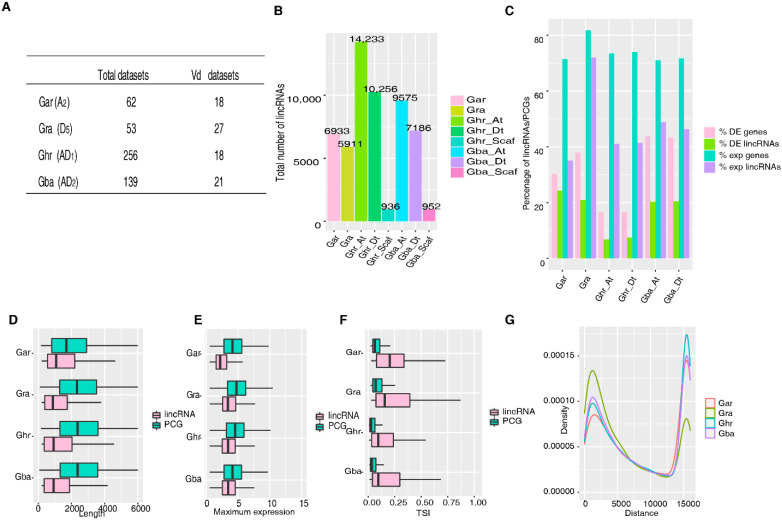
Features of the lincRNAs identified in diploid and allotetraploid cotton species. (**A**) The number of datasets used in identification of lincRNAs in diploid (Gar and Gra) and allotetraploid (Ghr and Gba) cotton species. *Vd*, *Verticillium dahliae*. (**B**) The number of lincRNAs identified in diploid (Gar and Gra) and the A_t_ and D_t_ subgenomes of the two allotetraploid cotton species (Ghr and Gba). (**C**) The percentage of the expressed protein coding genes (exp genes), the expressed lincRNAs (exp lincRNAs), the differentially expressed protein coding genes (DE genes), and the differentially expressed lincRNAs (DE lincRNAs) in the two diploids (Gar and Gra) and the A_t_ and D_t_ subgenomes of the two allotetraploid cotton species (Ghr and Gba). DE genes and DE lincRNAs were defined based on the datasets from *Vd*-infection experiments with the criteria of q-value < 0.05 and |log2(FC)| ≥ 1, using relatively highly expressed protein coding genes and lincRNAs (those with TPM < 0.5 in at least one sample were not considered). (**D**) The length of lincRNAs and protein coding genes (PCGs) in the four cotton species. (**E**) The maximum expression level (log2 transformed) of lincRNAs and PCGs in the four cotton species. (**F**) Distribution of the tissue specificity index (TSI, ranging from 0 to 1) of lincRNAs and PCGs in the four cotton species. Zero represents broadly expressed and one represents specific expression. (**G**) Distribution of the distance between lincRNAs and their nearest neighboring PCGs in the four cotton species.

**Figure 2 ijms-23-08594-f002:**
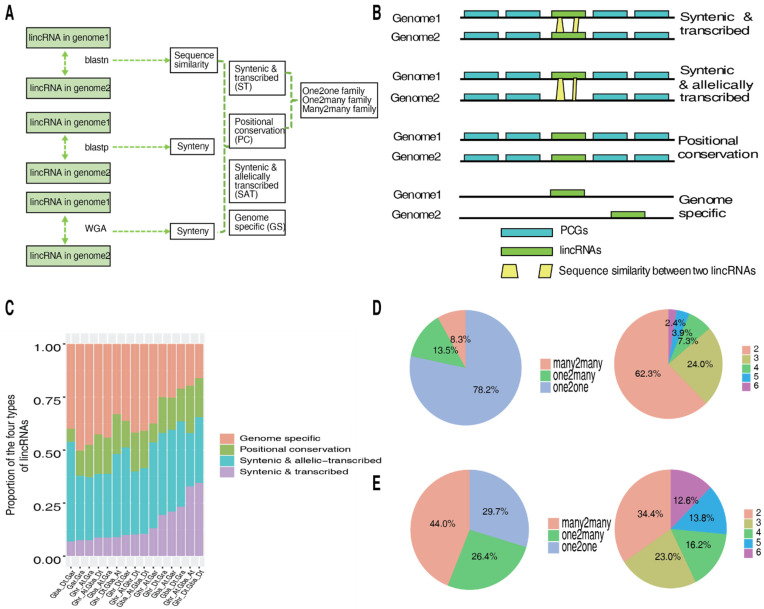
Conservation of lincRNAs in the diploid and allotetraploid cotton species. (**A**) The pipeline used in identification and classification of homologous lincRNAs. Homologous lincRNAs in different species were identified by three complementary approaches: sequence similarity by blastn, syntenic relationship by blastp (using flanking protein-coding genes or PCGs), and whole-genome alignment (WGA) by Lastz. The syntenic and transcribed (ST) and positional conservation (PC) lincRNAs were separated into three families (one2one, one2many, and many2many) based on the number of homologs of a certain lincRNA in the individual species. (**B**) LincRNAs were classified into four types, i.e., ST, syntenic and allelically transcribed (SAT), PC, and genome-specific (GS), based on their genomic position and sequence similarity. (**C**) The proportion of the four types of lincRNAs in the 15 individual pairwise comparisons among the six (sub)genomes (A_t_ and D_t_ subgenomes of Ghr and Gba as well as Gar and Gra). (**D**) Percentage of the ST lincRNAs assigned to the one2one, one2many, and many2many families (left pie), and percentage of the ST lincRNAs conserved in different number (from 2 to 6) of (sub)genomes (right pie). (**E**) Percentage of the PC lincRNAs assigned to the one2one, one2many, and many2many families (left pie), and percentage of the PC lincRNAs conserved in different number (from 2 to 6) of (sub)genomes (right pie).

**Figure 3 ijms-23-08594-f003:**
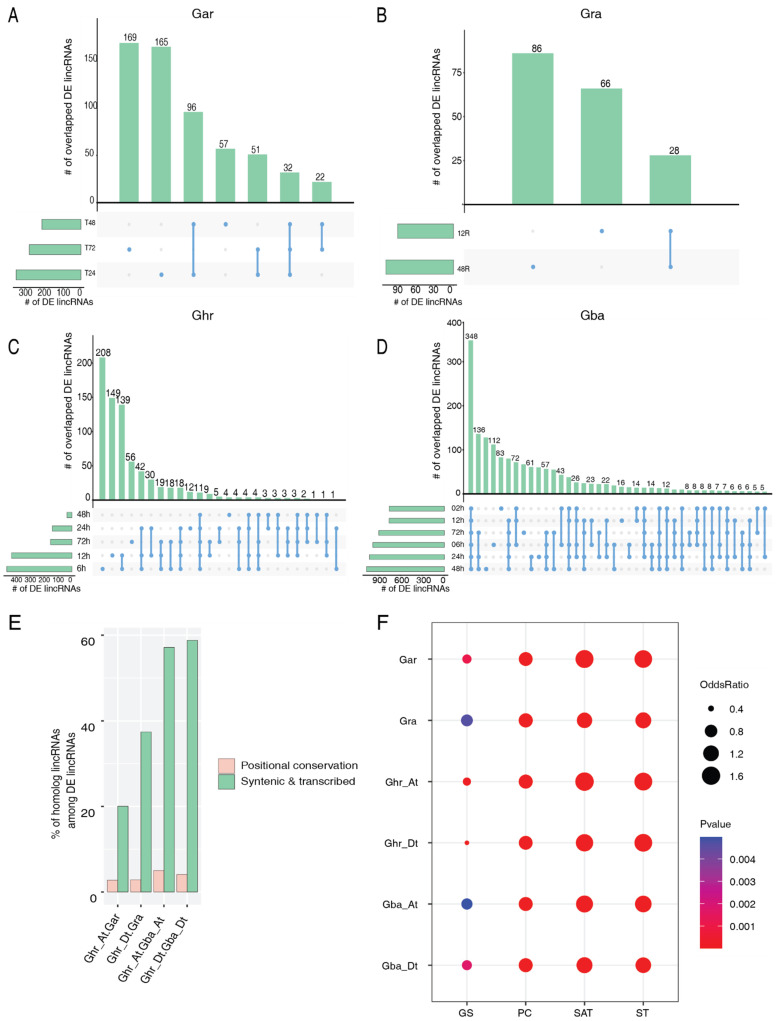
LincRNAs responding to *Vd* infection and their conservation. (**A**) The number of differentially expressed *Vd*-responsive lincRNAs at different time points in *G. arboreum* (Gar). Horizontal black bars represent the total number of differentially expressed *Vd*-responsive lincRNAs at the corresponding time points (treatment vs. control). The vertical black bars represent the number of differentially expressed *Vd*-responsive lincRNAs unique to a single time point (denoted by a black dot) or common to two or three time points (denoted by black lines linking the corresponding time points). This notation also applies to B-D. T24, T48, and T72 represent 24 h, 48 h, and 72 h after *Vd* infection, respectively. (**B**) The number of differentially expressed *Vd*-responsive lincRNAs at different time points in *G. raimondii* (Gra). The numbers 12 and 48 represent 12 h and 48 h after *Vd* infection, respectively. R: root. (**C**) The number of differentially expressed *Vd*-responsive lincRNAs at different time points in *G. hirsutum* (Ghr). h: hours post *Vd* infection. (**D**) The number of differentially expressed *Vd*-responsive lincRNAs at different time points in *G. barbadense* (Gba). h: hours post *Vd* infection. (**E**) Comparison of the percentage of *Vd*-responsive ST and PC lincRNAs in the four pairs of homologous (sub)genomes. (**F**) Association between *Vd* responsiveness and conservation of lincRNAs in the six (sub)genomes. ST: syntenic and transcribed; SAT: syntenic and allelically transcribed; PC: positional conservation; GS: genome-specific.

**Figure 4 ijms-23-08594-f004:**
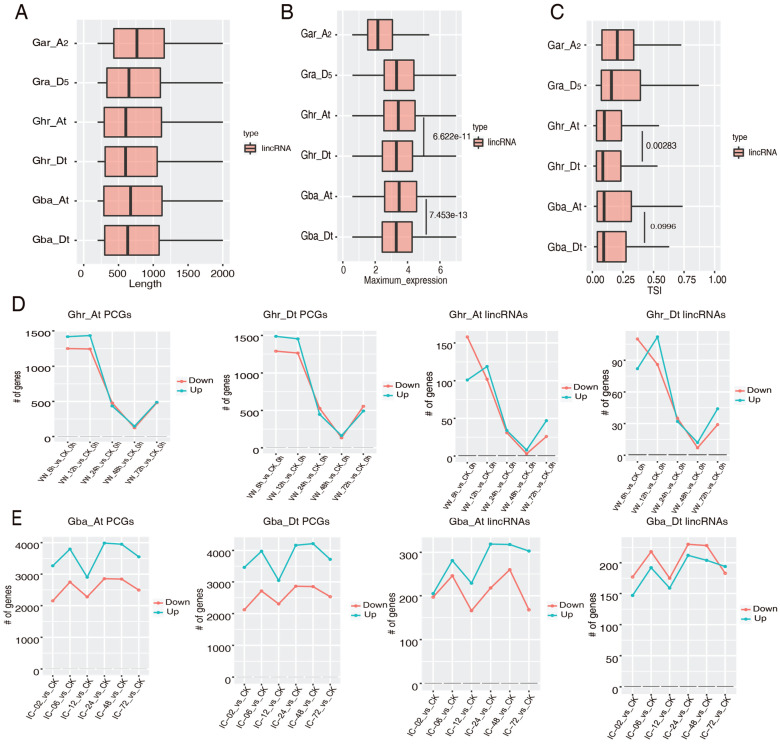
Subgenome dominance of the *Verticillium dahliae* (*Vd*)-responsive lincRNAs. (**A**) Comparison of the length of the *Vd*-responsive lincRNAs from the six cotton (sub)genomes. (**B**) Comparison of the maximum expression level (log2 transformed) of the *Vd*-responsive lincRNAs from the six cotton (sub)genomes. (**C**) Comparison of the expression specificity of the *Vd*-responsive lincRNAs from the six cotton (sub)genomes. (**D**) Comparison of the number of differentially expressed protein coding genes (PCGs) and lincRNAs in the *Vd*-infected samples in the two subgenomes of Ghr. (**E**) Comparison of the number of differentially expressed PCGs and lincRNAs in the *Vd*-infected samples in the two subgenomes of Gba.

**Figure 5 ijms-23-08594-f005:**
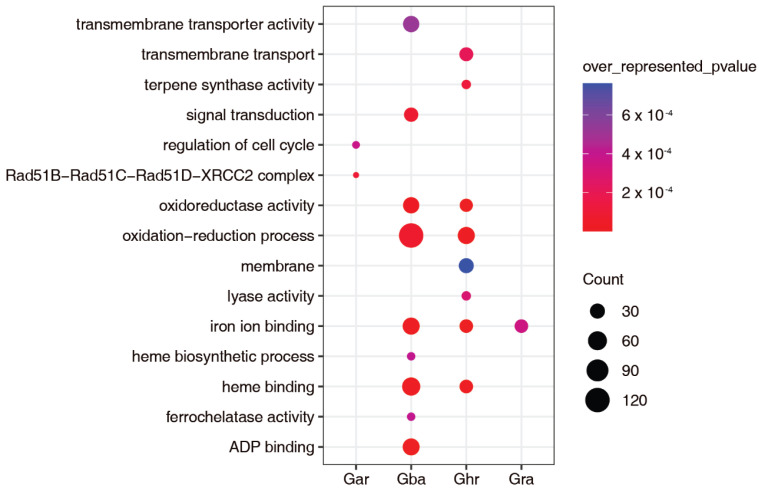
GO enrichment of the cis targets of the differentially expressed lincRNAs responding to *Vd* infection.

**Table 1 ijms-23-08594-t001:** Conservation of the syntenic and transcribed lincRNAs of the one2one family.

Type	Gar ^§^	Gra	Ghr_A_t_	Ghr_D_t_	Gba_A_t_	Gba_D_t_	No. of lincRNAs (%)
1a	√	√	√	√	√	√	48 (0.7)
1b	√	√	Lost in one to all four subgenomes				214 (3.1)
2a	√	x	√	√ or x	√	√ or x	577 (8.3)
2b	√	x	√	√ or x	x	√ or x	375 (5.4)
2c	√	x	x	√ or x	√	√ or x	233 (3.3)
2d	√	x	x	√ or x	x	√ or x	164 (2.4)
3a	x	√	√ or x	√	√ or x	√	505 (7.3)
3b	x	√	√ or x	√	√ or x	x	397 (5.7)
3c	x	√	√ or x	x	√ or x	√	262 (3.8)
3d	x	√	√ or x	x	√ or x	x	233 (3.3)
4	x	x	Presence in two to all four subgenomes				3954 (56.8)

^§^ √ and x represent presence and absence of homologous lincRNAs. respectively.

**Table 2 ijms-23-08594-t002:** Family classification of the stringently selected *Vd*-responsive lincRNAs.

Species	*Vd* Response	No. of DE lincRNAs	Unique to the Species ^§^	One2One	One2Many	Many2Many
Gar	Up	2	1 (50.0)	0	0	1 (50.0)
	Down	6	2 (33.3)	3 (50.0)	0	1 (16.7)
Gra	Up	28	14 (50.0)	5 (17.9)	4 (14.3)	5 (17.9)
	Down	19	12 (63.2)	5 (26.3)	0	2 (10.5)
Ghr	Up	33	20 (60.6)	10 (30.3)	2 (6.1)	1 (3.0)
	Down	48	23 (47.9)	13 (27.1)	4 (8.3)	8 (16.7)
Gba	Up	246	118 (48.0)	78 (31.7)	18 (7.3)	32 (13.0)
	Down	134	63 (47.0)	41 (30.6)	18 (13.4)	12 (9.0)

^§^ The numbers in parentheses represent percentage.

## Data Availability

All data are available in the manuscripts or the Appendix A.

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
