# Peer review of "The Conservation of Long Intergenic Non-Coding RNAs and Their Response to Verticillium dahliae Infection in Cotton"

_ijms, 2022, doi:10.3390/ijms23158594_

Round 1
Reviewer 1 Report
Authors submitted manuscript entitled “The conservation of long intergenic non-coding RNAs and their response to Verticillium dahliae infection in cotton” to IJMS. In this study they utilized publicly available RNA-seq datasets from different online sources, and several lincRNAs, in two cultivated allotetraploid cotton species, and two extant diploid progenitors of the allotetraploid cotton. This study reports good results and have utlized several bioinformatic approaches which can be beneficial for future studies to reveal new lincRNAs in cotton as well as other plant species. This study designed well but there are few sections of the manuscript needs to be substantially improved. Moreover, line numbering was missing in main file, so it was difficult to mention the exact location of mistakes/shortcomings.
There are numerous sentence, grammatical and typo mistakes. Moreover, many spacing issues were found. Please revise the manuscript carefully. It will be evaluated in second round strictly.
Figure 3D need some improvements. Figure is not very clear.
Figure 5 also need improvements.
Supplementary Figure 2 is provided two times? Keep only single Supplementary Figure 2. Moreover, take Supplementary Figure 1 to the same word file containing other supplementary figures.
Discussion section is weak. Please discuss your results in accordance with the findings and previously published literature. Moreover, draw conclusions based on the key findings and insert some sentences about the future possible applications of this study. The present discussion is very weak and not considerable.
I recommend drawing a flow chart for methodology used and put it in supplementary file. Try to elaborate each step followed to make it reader friendly. Like in Figure 2A, authors tried to explain the methodology, but it is not very reader friendly. Please make it comprehensive or remove this figure from here and insert a detailed figure in supplementary file.
Author Response
Response to Reviewer 1’s comments
Authors submitted manuscript entitled “The conservation of long intergenic non-coding RNAs and their response to Verticillium dahliae infection in cotton” to IJMS. In this study they utilized publicly available RNA-seq datasets from different online sources, and several lincRNAs, in two cultivated allotetraploid cotton species, and two extant diploid progenitors of the allotetraploid cotton. This study reports good results and have utlized several bioinformatic approaches which can be beneficial for future studies to reveal new lincRNAs in cotton as well as other plant species. This study designed well but there are few sections of the manuscript needs to be substantially improved. Moreover, line numbering was missing in main file, so it was difficult to mention the exact location of mistakes/shortcomings.
There are numerous sentence, grammatical and typo mistakes. Moreover, many spacing issues were found. Please revise the manuscript carefully. It will be evaluated in second round strictly.
Response: Thanks for your positive comments on our work. We have carefully gone through the manuscript and corrected as many as possible mistakes. Please see other responses below for the modifications that have been made to the manuscript based on your constructive suggestions and comments. We also have added line numbers in the revised manuscript.
Figure 3D need some improvements. Figure is not very clear.
Response: We have removed some of the numbers showing on top of the black bars to make the remaining numbers readable. Hope the removing of numbers does not affect interpretation of the Figure. We have also changed the style of all Figures.
Figure 5 also need improvements.
Response: We have changed the font size and presenting style to make the Fig looking not so crowd.
Supplementary Figure 2 is provided two times? Keep only single Supplementary Figure 2. Moreover, take Supplementary Figure 1 to the same word file containing other supplementary figures.
Response: Our apology for the duplication of Suppl Fig.2. We have added all Supplemental Figures (1 to 5) in a single word file. Supplemental Figure 1 is newly added to show the overall flowchart of the approaches used in this study for the major analyses as you suggested. The original Supplemental Figures 1 to 4 have been relabelled as Supplemental Figure 2 to 5, respectively.
Discussion section is weak. Please discuss your results in accordance with the findings and previously published literature. Moreover, draw conclusions based on the key findings and insert some sentences about the future possible applications of this study. The present discussion is very weak and not considerable.
Response: The Discussion was carefully crafted to achieve what the Reviewer suggested. We tried not to make overstatements and just to draw conclusions based on what we observed. At the end, we have added a couple of sentences regarding the pipeline we used in the study and the limitation of the study. The main findings and outcomes that can be further explored in the future (mainly the good candidate Vd-responsive lincRNAs) are summarised in the Conclusions.
I recommend drawing a flow chart for methodology used and put it in supplementary file. Try to elaborate each step followed to make it reader friendly. Like in Figure 2A, authors tried to explain the methodology, but it is not very reader friendly. Please make it comprehensive or remove this figure from here and insert a detailed figure in supplementary file.
Response: Following your constructive suggestion, we have added an overall flowchart in Suppl. Fig 1 to show the methodology used in the study, and modified Fig 2A accordingly. We hope that the flowchart together with the details presented in the Methods section would be good enough for others who are interested in using the method in similar investigation.
Reviewer 2 Report
In the MS, the authors investigated lincRNA in the available transcriptomic data in Cotton, comparing the differences between cultivated species (diploids and tetraploids). Further, the lincRNA correlation with Verticilium dahliae response have been investigated.
The experimental design is well described and robust, the text is well developed and the results clearly presented.
After a round of minor revisions (please see the attached file), I think the MS will be ready for Editors decision.

Author Response
Response to Reviewer 1’s comments
In the MS, the authors investigated lincRNA in the available transcriptomic data in Cotton, comparing the differences between cultivated species (diploids and tetraploids). Further, the lincRNA correlation with Verticilium dahliae response have been investigated.
The experimental design is well described and robust, the text is well developed and the results clearly presented.
Response: Thanks for the positive comments on the work.
After a round of minor revisions (please see the attached file), I think the MS will be ready for Editors decision.
Response: We have taken account of all your comments marked in the annotated pdf file by rewording sentences or correcting typos. Regarding the unpublished PacBio SMRT sequencing data, we have changed “unpublished data” to “to be published separately”, as the data are the major focus of another study but not this study. We apology for the messy Figure 5, but it could be caused by file conversion when generating the pdf file.
Round 2
Reviewer 1 Report
Authors have addressed all the comments.